# The initial charge separation step in oxygenic photosynthesis

Yusuke Yoneda [1,2,5], Eric A. Arsenault [1,2,3], Shiun-Jr Yang[1,2], Kaydren Orcutt [1,2], Masakazu Iwai [2,4] & Graham R. Fleming [1,2,3 ✉]

Photosystem II is crucial for life on Earth as it provides oxygen as a result of photoinduced electron transfer and water splitting reactions. The excited state dynamics of the photosystem II-reaction center (PSII-RC) has been a matter of vivid debate because the absorption spectra of the embedded chromophores significantly overlap and hence it is extremely difficult to distinguish transients. Here, we report the two-dimensional electronic-vibrational spectroscopic study of the PSII-RC. The simultaneous resolution along both the visible excitation and infrared detection axis is crucial in allowing for the character of the excitonic states and interplay between them to be clearly distinguished. In particular, this work demonstrates that the mixed exciton-charge transfer state, previously proposed to be responsible for the far-red light operation of photosynthesis, is characterized by the $Chl_{D1}^+$Phe radical pair and can be directly prepared upon photoexcitation. Further, we find that the initial electron acceptor in the PSII-RC is Phe, rather than $P_{D1}$, regardless of excitation wavelength.

[1] Department of Chemistry, University of California, Berkeley, CA 94720, United States. [2] Molecular Biophysics and Integrated Bioimaging Division, Lawrence Berkeley National Laboratory, Berkeley, CA 94720, United States. [3] Kavli Energy Nanoscience Institute at Berkeley, Berkeley, CA 94720, United States. [4] Department of Plant and Microbial Biology, University of California, Berkeley, CA 94720, United States. [5]Present address: Research Center of Integrative Molecular Systems, Institute for Molecular Science, National Institute of Natural Sciences, Okazaki, Aichi 444-8585, Japan. ✉email: grfleming@lbl.gov

Photosynthesis, the green engine of life on Earth, produces molecular oxygen by using the light-driven water-plasto-quinone oxidoreductase enzyme known as photosystem II[1–3]. The photosystem II-reaction center (PSII-RC) is one of the smallest photosynthetic components which can undergo charge separation (CS) and thus is an ideal model system to investigate the underlying mechanism of the initial light-energy conversion process of photosynthesis[4–6]. The PSII-RC consists of six pigments as central cofactors—two special pair chlorophylls ($P_{D1}$ and $P_{D2}$), two accessory chlorophylls ($Chl_{D1}$ and $Chl_{D2}$), and two pheophytins ($Phe_{D1}$ and $Phe_{D2}$)—arranged in a quasi-symmetric geometry (Fig. 1a)[7,8]. These six molecules are generally referred to as RC pigments. In addition, there are two peripheral antenna Chls which are denoted as $Chlz_{D1}$ and $Chlz_{D2}$. Despite the similarity of the pigment arrangement in the D1 and D2 branches, electron transfer only proceeds along the D1 pigments. The specifics of how CS proceeds in the PSII-RC is, however, a matter of vivid debate. In particular, there remains a long-standing discussion concerned with whether the initial electron acceptor is $P_{D1}$[9,10] or $Phe_{D1}$[11–13], i.e., whether the initial radical pair is ($P_{D2}^+P_{D1}^-$) or ($Chl_{D1}^+Phe_{D1}^-$). The uncertainty here is a consequence of the many closely spaced excitonic states arising from pigment-pigment interactions in the PSII-RC such that no observable structure is present even in the electronic linear absorption spectrum at cryogenic temperatures[14–16].

To this end, the excited state dynamics of the PSII-RC has been the focus of extensive spectroscopic interest spanning over three decades. These works have included time-resolved fluorescence[17,18], transient absorption[9,10,13,19–21], optical photon-echo[12], visible pump-mid infrared (IR) probe[11], and two-dimensional electronic spectroscopy[14,22–24] studies. While electronic spectroscopies acutely suffer from a lack of spectral resolution in regard to the PSII-RC, the implementation of mid-IR spectroscopy has proven to be highly advantageous in addressing issues related to spectral congestion[25–28]. In particular, the keto and ester CO stretching modes of Chl and Phe show unique signatures in the mid-IR region depending on the local protein environment, electronic structure, and ionic states[11,29–33]. Additionally, the amide I modes of the backbone protein can be used as sensitive reporters for the electron transfer[11,31]. This was notably demonstrated by Groot et al. in a visible pump-mid IR probe study of

the PSII-RC where it was suggested that the initial electron acceptor was Phe based on its distinguishing vibrational structure[11]. However, spectral resolution only along the detection axis was not sufficient to disentangle the distinct excitonic contributions and dynamics or definitively assign the initial electron acceptor.

Many theoretical models have been developed in order to aid in experimental interpretation and to elucidate the nature of the electronic states at different absorption wavelengths[34–37]. Particularly, Stark spectroscopy suggests that the absorption spectrum of PSII is not characterized by purely excitonic states, rather it is composed of mixed exciton-charge transfer (CT) states possibly including contributions from $(Chl_{D1}^{\delta+}Phe_{D1}^{\delta-})^*$ and $(P_{D2}^{\delta+}P_{D1}^{\delta})^*$[38]. In an attempt to model this, one of the most sophisticated exciton models of the PSII-RC takes into account eight pigments—the six RC and two peripheral pigments—and one CS state[34]. Even in this model, there was uncertainty as to the character of the initial CS state because both $P_{D2}^+P_{D1}^-$ and $Chl_{D1}^+Phe_{D1}^-$ gave reasonable fits to the data with the former yielding slightly better agreement to experimental data considered. It is important to note here that the experimental data was, however, entirely from electronic spectroscopies.

While uncertainty surrounds the identity of the states involved in exciton-CT mixing in the PSII-RC, studies have suggested that the mixed CT states are responsible for the far-red excitation of PSII[39–41]. Although the absorption of the PSII-RC and the required redox potential of water oxidation were believed to be located below 690 nm, it was demonstrated that PSII can be operated by the far-red light beyond 690 nm (exhibiting activities including oxygen evolution)[39,42]. Additionally, recent EPR experimental[40] and QM/MM theoretical[41] studies suggest that the far-red light excitation of PSII involves a lower-lying CT state with a hole localized on $Chl_{D1}$ rather than $P_{D2}$. However, just as spectral congestion obscures the assignment of the initial electron acceptor, the character of these mixed CT states remains undetermined.

Compared to the previously mentioned techniques, the emerging method of two-dimensional electronic-vibrational (2DEV) spectroscopy, which correlates electronic excitation and mid-IR detection[43–48], has the potential to overcome the challenges associated with congested electronic spectra. In particular, the

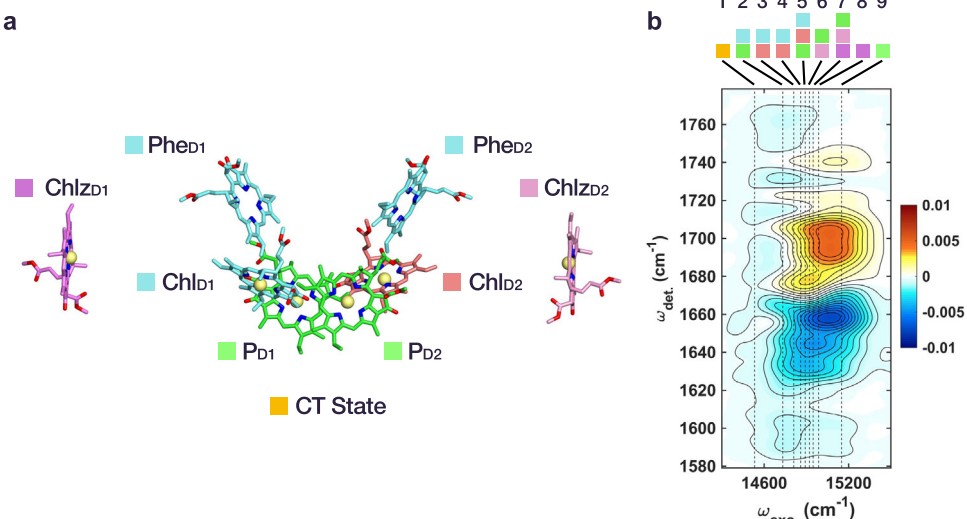

**Fig. 1 Structure and 2DEV spectrum of the PSII-RC. a** Pigment arrangement of the PSII-RC depicted based on the crystal structure (3WU2) reported by Umena et al.[8]. **b** 2DEV spectrum of the PSII-RC at 180 fs. Positive contours (red/yellow) indicate ground state bleach (GSB) features and negative contours (blue) indicate photoinduced absorption (PIA) features. The vertical dotted lines show the zero phonon exciton transition energies based on the model by Novoderezhkin et al.[34]. Contour levels are drawn in 5% intervals. Colored squares on the top indicate the dominant pigments participating in each excitonic state as labeled in **a**.

simultaneous spectral resolution along both the visible excitation and IR detection axis has been shown to enable the clear assignment of transient species[44–48]. In this study, we investigated the excited state dynamics of the PSII-RC via 2DEV spectroscopy. Both highly excitation frequency-dependent spectral structure and dynamics were clearly resolved. This allowed for a broad analysis of the excitonic composition of the PSII-RC and direct insight into the involvement of mixed exciton-CT states found to be directly prepared upon photoexcitation. Further, the spectra facilitated an assignment of the initial electron acceptor and enabled the excitation energy transfer (EET) and electron transfer pathways initiated by peripheral antenna excitation or RC pigments excitation to be disentangled.

## Results and discussion

**General insights from the 2DEV spectra and IR band assignments.** Figure 1b shows the 2DEV spectrum of the PSII-RC 180 fs after photoexcitation. Of note is the significant excitation frequency ($\omega_{exc.}$)-dependence of the vibrationally resolved structure along the detection axis ($\omega_{det.}$) which, as we will demonstrate, allows for an excitonic state-specific analysis of the spectra with high frequency resolution (i.e., vibrationally resolved excitonic structure). For example, absorption from newly formed species (photoinduced absorptions, PIA), spanning $\omega_{det.} = 1710–1760$ cm$^{-1}$ were seen to be more intense for the lower-lying excitonic states. Other strong indications of this $\omega_{exc.}$-dependent behavior were observed in the ground state bleach (GSB) region spanning $\omega_{det.} = 1680–1710$ cm$^{-1}$ and the PIAs at $\omega_{det.} = 1620–1670$ cm$^{-1}$. These three regions are of particular interest because, here, vibrational modes belonging to both the neutral and ionic forms of Chl and Phe can be clearly distinguished—thus serving as sensitive markers for the EET and CT steps leading to CS as well as the character of the excitonic states.

The vibrational structure of the PSII-RC is not only highly $\omega_{exc.}$-dependent, but also shows a significant time-dependence. Therefore, our assignments will be based on the vibrational structure at specific $\omega_{exc.}$ corresponding to the energies of exciton 2 (14,690 cm$^{-1}$) and exciton 8 (14,940 cm$^{-1}$) in the model by Novoderezhkin et al.[34], which covers the relevant pigments along the D1 branch, and at either early or later waiting times (Fig. 2).

Generally, the GSB observed at $\omega_{det.} = 1680–1710$ cm$^{-1}$ is assigned to the keto CO stretching mode of Chl/Phe[29,31,32]. On the electronic ground state, the frequency of this keto mode depends on the polarity of the environment and the presence of hydrogen bonding from surrounding media (the larger the polarity, or the stronger the hydrogen bond, the lower the frequency of the keto mode). Thus, the GSB can be used to broadly distinguish pigment contributions (further discussed in the next section). For example, in Fig. 2, it is apparent at early waiting times that the GSB band of exciton 8 shows much more signal amplitude at 1680–1700 cm$^{-1}$ compared to that of the exciton 2. This is in line with a light-induced FTIR difference spectroscopic study which reported that Chlz shows a GSB at 1684 cm$^{-1}$ [31], whereas P and Phe exhibit higher and lower frequency GSBs at 1704 cm$^{-1}$ and 1677 cm$^{-1}$, respectively[29,31,32]. In addition, the GSB frequency of triplet-carrying Chl, either Chl$_{D1}$ or Chl$_{D2}$, has been reported to be at 1670 cm$^{-1}$ [31,32].

On the electronically excited state, the keto modes of Chl and Phe exhibit redshifted absorption[11,49]. For example, in THF, the keto stretching mode in the previously measured Chl*/Chl difference spectrum was seen to shift from 1695 cm$^{-1}$ to 1660 cm$^{-1}$ [11]. Correspondingly, the negative signal at $\omega_{det.} = 1620–1670$ cm$^{-1}$ in both exciton 2 and 8 is broadly assigned to the excited state absorption (ESA) of the keto modes of Chl and Phe. At later waiting times, however, there is a notable evolution in the vibrational

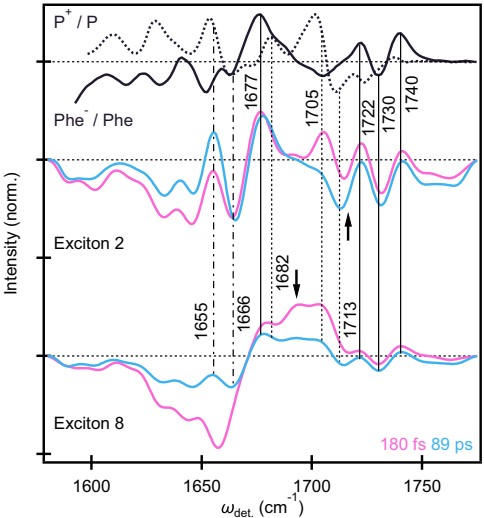

**Fig. 2 Exciton-specific vibrational structure and IR assignments.** Slices of 2DEV spectrum at $\omega_{exc.} = 14,690$ cm$^{-1}$ and $\omega_{exc.} = 14,940$ cm$^{-1}$, corresponding to the energies of exciton 2 and 8 at early (pink, 180 fs) and later (blue, 89 ps) waiting times. The difference absorption spectra of P$^+$/P (dotted line) and Phe$^-$/Phe (solid line) are shown above for comparison (where the signs have been reversed to match the convention of the 2DEV data). Vertical dotted (solid) lines indicate band assignments corresponding P$^+$/P (Phe$^-$/Phe) while dash-dotted lines distinguish more ambiguous assignments. The black arrow in exciton 2 marks the Chl$_{D1}^+$ mode at 1716 cm$^{-1}$ and in exciton 8 marks the Chlz$_{D1}$ ground state bleach. The P$^+$/P and Phe$^-$/Phe spectra are reproduced from Refs. 30 and 29 with permission.

structure of this region (Fig. 2). A clear GSB peak at 1655 cm$^{-1}$, overlapping with a broad ESA feature, appeared concomitantly with a new peak emerging at 1666 cm$^{-1}$. Both the P$^+$/P and Phe$^-$/Phe difference spectra exhibit GSB features in this region at frequencies of 1653–1655 cm$^{-1}$ and 1659 cm$^{-1}$ [29,31,32], respectively. Resonance Raman spectroscopy of PSII-RC shows no signal at 1640–1660 cm$^{-1}$, thus Groot et al. and Noguchi et al. suggest that the band at 1655 cm$^{-1}$ is assigned to the amide CO mode reflecting the CS at the RC, rather than keto stretching mode of Chl or Phe[11,31]. The band at 1666 cm$^{-1}$ is similar to both Phe$^-$/Phe and P$^+$/P showing signals at 1662 cm$^{-1}$ and 1663 cm$^{-1}$ [29,32], respectively, which has been suggested as a counterpart of the previously mentioned band[31]. A more definitive assignment is reserved for later discussion.

This leaves the remaining PIA region spanning 1710–1760 cm$^{-1}$. While the ester modes Chl* and Phe* fall in this region[11], they are known to be very weak and would be unlikely to account for the full intensity of the observed features. Further, assuming that this region is only composed of Chl* and Phe* ester modes would not account for the significant $\omega_{exc.}$-dependence clearly present in Fig. 1b. If this was the case, then this region should have a near-uniform intensity across excitons 3 through 7 which have similar pigment contributions and exciton transition dipole strengths[34], but this is clearly not so (Fig. 1b). As a result, contributions from Chl* and Phe* ester modes are likely small, which should leave this a relatively clear spectral window, yet, strong features are apparent in the 2DEV spectra. The Phe$^-$/Phe difference spectrum measured in PSII, however, shows characteristic signatures in this region, still related to the ester mode of chromophore itself or surrounding amino acid residue, with strong absorptions at 1722 cm$^{-1}$, 1730 cm$^{-1}$, and 1739 cm$^{-1}$ (Fig. 2)[29,32]. The corresponding peaks in the 2DEV spectrum (at 1722 cm$^{-1}$, 1730 cm$^{-1}$, and 1740 cm$^{-1}$), apparent at early waiting times for exciton 2 and emerging later for exciton 8,

are therefore assigned to Phe. It should be noted that exciton 8 does show a small negative signal around 1730 cm$^{-1}$ and a positive band at 1740 cm$^{-1}$ immediately after photoexcitation, despite being characterized by Chlz$_{D1}$. We attribute these signals to either small contributions from the ester ESA or some degree of overlap between excitonic bands, as these slices only represent the calculated zero phonon transitions and the actual absorption has finite bandwidth.

Further characteristic of the Chl $a$ cation is a significantly blueshifted keto stretch, to 1718 cm$^{-1}$, (on the order of 25 cm$^{-1}$) versus neutral Chl $a$ in THF[33]. At early waiting times in exciton 2, for example, a peak is oberved at 1716 cm$^{-1}$ which we assign to Chl$_{D1}^+$. However, at later waiting times, this peak noticeably redshifts to 1713 cm$^{-1}$, toward agreement with the characteristic P$^+$ absorption at 1711 cm$^{-1}$. This dynamical behavior will be the focus of later discussion.

It should be noted that the steady state spectrum of P$_{D2}^+$P$_{D1}^-$ is not measurable for a comparison because this species expected to be short-lived (if it indeed exists as an intermediate)[21,41]. We therefore estimate the characteristic bands of P$_{D2}^+$P$_{D1}^-$ based on the assumption that cation and anion formation in P$_{D2}$P$_{D1}$ will exhibit similar spectral shifts to monomeric Chl because the charges should be localized on P$_{D2}$ and P$_{D1}$, respectively. The keto CO of the Chl generally red shifts compared to the ground state species[50], however, this frequency falls in a congested region of the spectrum for the PSII-RC. On the other hand, the keto CO of monomeric Chl shows a ~25 cm$^{-1}$ blue shift upon cation formation[33]. Given the main GSB peak of P is at 1701 cm$^{-1}$, we can expect that the characteristic band of P$_{D2}^+$P$_{D1}^-$ should appear at ~1726 cm$^{-1}$. However, we only observe clear signatures of Phe bands (and associated GSBs) at 1730 cm$^{-1}$ (and 1722 and 1740 cm$^{-1}$) across the entire excitation axis.

To summarize, the significant markers tracking CS in this study are as follows (Table 1): Phe (1722 cm$^{-1}$, 1730 cm$^{-1}$, and 1740 cm$^{-1}$), Chl$_{D1}^+$ (at early waiting times: 1716 cm$^{-1}$), P$^+$ (at later waiting times: 1713 cm$^{-1}$), and the GSB of the amide CO bands at 1655 cm$^{-1}$ and its up-shifted counterpart at 1666 cm$^{-1}$. As the excitonic states of the PSII-RC are delocalized over several chromophores, we focus our discussion below on the CS markers rather than GSB and ESA features spanning 1680–1710 cm$^{-1}$ and 1620–1670 cm$^{-1}$, respectively, which reflect the relaxation of delocalized excitonic states.

**Excitonic composition and charge transfer character.** Following the vibrational assignments, we focus on a comparison of the vibrational structure at specific excitonic energies based on the model by Novoderezhkin et al.[34], in order to understand the character of the excitonic states and degree of CT mixing. Fig. 3a shows the vibrational structure corresponding to exciton 1, 2, 5, and 8 at an early waiting time. We note again that the exciton energies discussed thus far are zero phonon lines (shown in Fig. 1b). However, it has been reported that the actual absorption of the CT state shows a significant blue shift (~5 nm) as a result

of coupling to low-frequency phonons in the environment, compared to other excitonic bands (1–2 nm)[34]. Thus, to investigate the CT state specifically, the 2DEV signal corresponding exciton 1 as shown in Fig. 3a was integrated in the range $\omega_{exc.}$ = 14,500–14,650 cm$^{-1}$.

At early time, the exciton 1 signal, formed directly upon photoexcitation, shows clear structure corresponding to Phe (1722 cm$^{-1}$, 1730 cm$^{-1}$, and 1740 cm$^{-1}$), Chl$_{D1}^+$ (1716 cm$^{-1}$). In addition, the frequency of the GSB band around 1675 cm$^{-1}$ for exciton 1 is lower than other excitonic states. This is in agreement with the previous reports that the GSB frequencies in this range of Chl$_{D1}$ (1670 cm$^{-1}$) and Phe (1677 cm$^{-1}$) are redshifted compared to those of P (1682 cm$^{-1}$) and Chlz (1684 cm$^{-1}$)[31,32]. Furthermore, the amide CO bands reflecting CS at 1655 cm$^{-1}$ and 1666 cm$^{-1}$ show clear structure compared to the other excitonic states, highlighting the significant CT character of exciton 1 state. The characteristic P$^+$ signal (1713 cm$^{-1}$) appears at later waiting times and is accompanied by evolution at both of the aforementioned band positions (Fig. 4)—collectively indicating a conspicuous lack of initial contributions from P$_{D2}^+$P$_{D1}^-$.

The lack of P$_{D2}^+$P$_{D1}^-$ is in contrast to several previous spectroscopic studies that suggested there are two CS pathways in the PSII-RC[21,22,24,38]. However, the resolution afforded by both the visible excitation and IR detection dimensions of the 2DEV spectrum lead in particular to the vibrational characterization of exciton 1—providing direct evidence that the initial CT state in the PSII-RC is characterized by Chl$_{D1}^+$Phe$^-$ rather than P$_{D2}^+$P$_{D1}^-$ (Fig. 3b). Such a result is consistent with a recent QM/MM calculation, utilizing range-separated TD-DFT theory and the coupled-cluster theory with single and double excitations (CCSD), which proposed that the lowest CT state was Chl$_{D1}^+$Phe$^-$[41]. Recent theoretical studies suggest that the lowest CT state among the RC pigments is composed of P$^+$Phe$^-$[51,52] and that state, which has very low oscillator strength, can be directly excited by far-red light (in the red tail of, or beyond our laser spectrum)[52]. Our spectra show similar frequencies for Chl$_{D1}^+$ and P$^+$, thus it is possible that there is a small contribution from P$^+$ to the signal even at early time. It is clear, however, that the majority of the initial signal at 1716 cm$^{-1}$ and 1677 cm$^{-1}$ arises from Chl$_{D1}$ because of the significant oscillator strength of Chl$_{D1}^+$Pheo$^-$ transition[52] in addition to indicating that the initial electron acceptor is Phe. A previous transient IR study also suggested that the initial electron acceptor is Phe[11], however, this study relied on an extrinsic deconvolution of the vibrational spectrum as opposed to the intrinsic ability of 2DEV spectroscopy to separate excitonic and CT contributions along the $\omega_{exc.}$ dimension. This advantage of 2DEV spectroscopy is particularly useful in the characterization of the CT state which is only weakly optically allowed and can therefore be easily obscured in other spectroscopic methods.

Considering the other states, an analysis of the GSB features of exciton 2 and 8 characterize these excitons as predominantly composed of RC pigments in the active (D1) branch and of the peripheral Chlz$_{D1}$, respectively, which is consistent with the model put forth by Novoderezhkin et al. (Fig. 3b)[34]. These assignments also substantiate that Chl and Phe at different binding positions in the PSII-RC are indeed excited by different excitation frequencies—offering support for the importance of the protein environment in tuning the site energies of the embedded pigments[41].

Exciton 2 also notably displays characteristic Chl$_{D1}^+$ and Phe$^-$ signals at early waiting times (Fig. 3a). In comparison to exciton 5, which is mainly composed of RC pigments in addition to Chlz$_{D2}$ (Fig. 3b), these CT signatures in exciton 2 are markedly more pronounced. Here, we have chosen exciton 5 as a representative for the energetically intermediate excitonic states,

**Table 1 IR frequency assignments of the PSII-RC.**

| Frequency (cm$^{-1}$) | Assignment |
| --- | --- |
| 1655 | GSB of amide CO[11,31] |
| 1666 | PIA of amide CO[11,31] |
| 1677 | GSB of Phe[29,32] |
| 1682 | GSB of P[31,32] |
| 1705 | GSB of P[31,32] |
| 1713 | PIA of P$^+$[31,32] |
| 1722 | GSB of Phe or amide CO[29,32] |
| 1730 | PIA of Phe or amide CO[29,32] |
| 1740 | GSB of Phe or amide CO[29,32] |

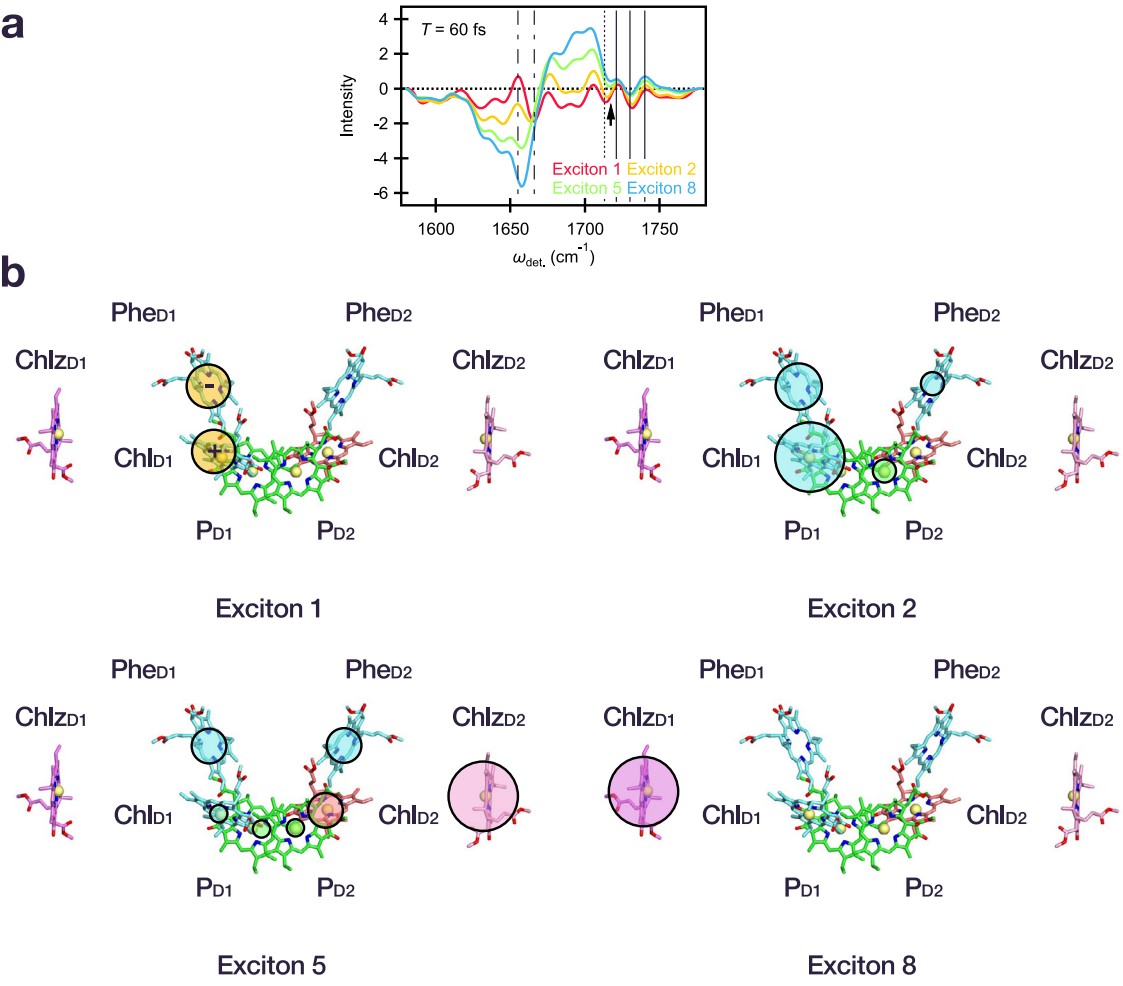

**Fig. 3 Assignment of excitonic composition and charge transfer character. a** Slice along $\omega_{det.}$ of the 2DEV spectrum corresponding to exciton 1 (red, integrated at $\omega_{exc.} = 14{,}500$–$14{,}650$ cm$^{-1}$), exciton 2 (yellow, $\omega_{exc.} = 14{,}690$ cm$^{-1}$), exciton 5 (green, $\omega_{exc.} = 14{,}850$ cm$^{-1}$), and exciton 8 (blue, $\omega_{exc.} =$ 14,940 cm$^{-1}$) at a waiting time of 60 fs. The vertical solid, dotted, and dash-dotted lines, as well as the black arrow follow the same convention as in Fig. 2. **b** Character of initial charge transfer state, exciton 1, along with the site contributions of excitons 2, 5, and 8 where the area of the shaded circles is proportional to the population of the corresponding sites based on the model of Novoderezhkin et al.[34]. For clarity, the slight, additional contributions from D1 pigments, nearly identical to the relative contributions of exciton 2, were omitted from exciton 1. Likewise, the charge transfer character present in excitons 2 and 5 was excluded for simplicity.

where there is congestion even in the 2DEV spectra. However, the vibrational structure is still telling in that the additional Chlz$_{D2}$ contributions of exciton 5 should be similar to those of Chlz$_{D1}$, which is indeed reflected in the fact that exciton 5 resembles a mixture of exciton 2 (mainly RC pigments) and exciton 8 (mainly composed of a peripheral pigment). This comparison highlights the enhanced CT character in exciton 2 versus exciton 5 at early waiting times which confirms the suggestion put forth in the model by Novoderezhkin et al.[34] that exciton 2 is responsible for initiating primary charge separation. Further, in the model, exciton 1 was taken to be characterized by a CT state which borrowed intensity from the neighboring state, exciton 2. This is in agreement with the close resemblance between the GSB and ESA (particularly below 1650 cm$^{-1}$ which is outside of the dominant window for the CS markers) structure of exciton 1 compared to that of exciton 2 (Fig. 3a) and signifies similar overall pigment contributions. This point is made even clearer by comparison of exciton 1 versus exciton 5 or 8 where there is little similarity in these regions. Correspondingly, this indicates that exciton 2 is characterized by a mixed exciton-CT state, rather than a purely excitonic state that rapidly evolves to the CT state. The mixed character between exciton 1 and 2 also offers a

mechanism through which rapid charge separation can be initiated in the RC.

**Charge separation dynamics**. To elucidate the dynamics, a global analysis of the data with sequential modeling was performed. We note that while the time constants represent a convolution of various processes, this method is able to holistically capture the spectral evolution along both frequency dimensions. Therefore, the analysis captures the $\omega_{exc.}$-dependent spectra and dynamics, and the latter can be largely disentangled via vibrational signatures as we will show. The two-dimensional-evolution associated difference spectra (2D-EADS) analysis (Fig. S2), which can be thought as the two-dimensional analog of EADS[53], required five components for a reasonable fit (35 fs, 1.3 ps, 6.3 ps, 41 ps, and a non-decaying offset component beyond 100 ps, the duration of the experiment). We note that the actual dynamics of the PSII-RC is not a simple sequential process as parallel and reversible processes are also expected[11,13]. Thus, we will discuss the dynamics based on the 2DEV slices rather than relying directly on the EADS.

Figure 4 contains exciton-specific slices through the actual 2DEV spectra along $\omega_{det.}$ at the earliest resolvable waiting time and

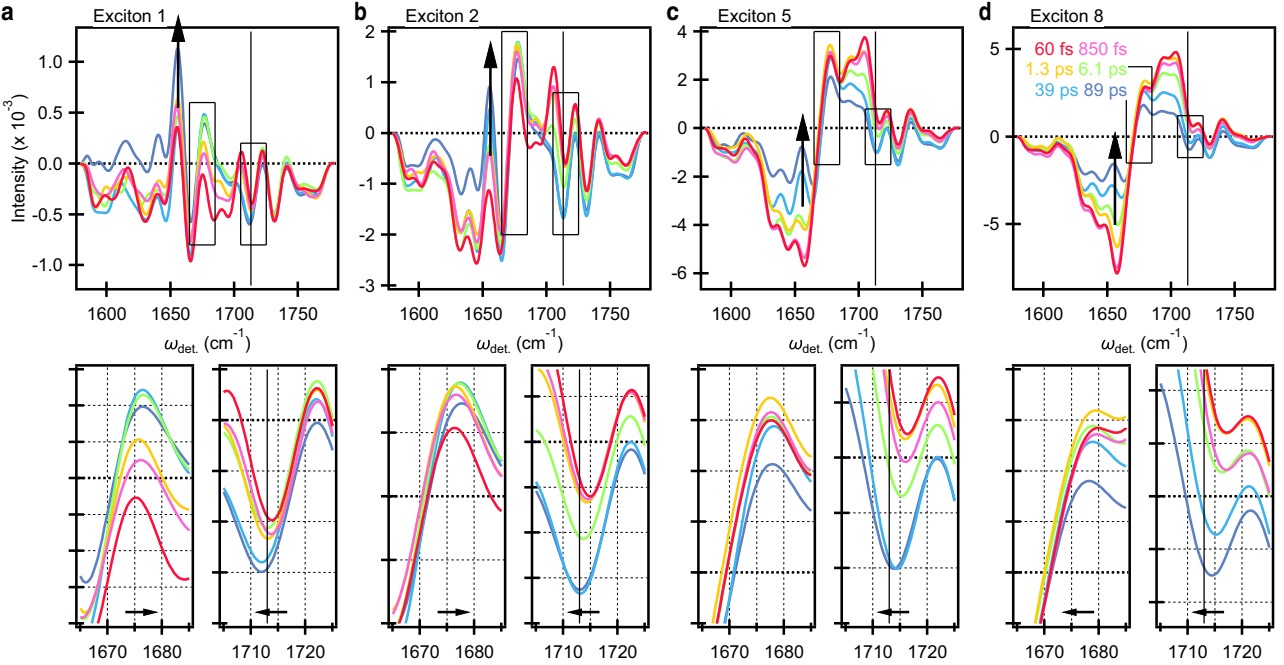

**Fig. 4 Dynamics of the PSII-RC.** The time-dependent evolution of 2DEV spectra corresponding to excitons **a** 1, **b** 2, **c** 5, and **d** 8. The energy ranges for $\omega_{exc.}$ are identical to those in Fig. 3. The waiting times are 60 fs (red), 850 fs (pink), 1.3 ps (yellow), 6.1 ps (light green), 39 ps (light blue), and 89 ps (blue). Bottom left and right panels show the range of $\omega_{det.} = 1665-1695\ cm^{-1}$ and $1705-1725\ cm^{-1}$, highlighting the shifting behavior of the GSB band of Chl and red-shifting behavior of the $Chl^+$ band.

at subsequent waiting times corresponding to each of the above mentioned time constants. The fastest component with a time constant of 35 fs is below the time resolution of our experimental system (~100 fs), and thus it can reflect a coherent artifact around time zero. Therefore, we will concentrate our discussion on the later time scales. Throughout, we focus our attention on excitons 2, 5, and 8 as these states have substantially more oscillator strength than exciton 1 and therefore will have a larger influence on the obtained time constants. The evolution associated with these time constants can be interpreted such that each spectrum (or slice) evolves into the next one with the associated time constant. For example, in exciton 2 (Fig. 4b), spectral evolution on the 1.3 ps timescale (comparison of the yellow and green slices in Fig. 4b), it exhibits growth at $1655\ cm^{-1}$, $1666\ cm^{-1}$, 1716, and $1730\ cm^{-1}$ while a slight shoulder begins to emerge in $1655-1666\ cm^{-1}$ region for exciton 5. This evolution is also accompanied by marked changes in the keto ESA structure. We assign the 1.3 ps timescale to progressive completion of CS, i.e., $(Chl_{D1}{}^{\delta+}Phe^{\delta})^* \longrightarrow Chl_{D1}{}^+Phe$ (more pronounced for exciton 2), convoluted with EET within the excitonic manifold (more pronounced for exciton 5) and an environmental response. This timescale also agrees with previous works which suggested that initial CS occurs with 600–800 fs[11] or 2–4 ps[13], among others which have reported multiexponential CS dynamics[21,24]. The distinction here is that the vibrational structure allows for a targeted assessment of the dynamical components for each of the states.

On an 6.3 ps timescale, both the $1657\ cm^{-1}$ and $1666\ cm^{-1}$ CS markers exhibit further evolution along with a distinct, progressive redshift in the band at $1716\ cm^{-1}$ to $1713\ cm^{-1}$ for excitons 1, 2, and 5. This component is similar to the previously reported timescale for $Chl_{D1}{}^+Phe \longrightarrow P^+Phe$ of 6 ps[11]. Additionally, in a previous light-induced FTIR difference spectroscopic study, it was proposed that the blue shift of the keto stretch of Chl cation is smaller for the charge delocalized dimeric Chl (~10 cm$^{-1}$ in the case of P680$^+$) compared to that of monomeric Chl (~30 cm$^{-1}$)[54]. Both experimental[54,55] and

theoretical[56,57] efforts further support that the P680 cation is partially delocalized over the $P_{D1}$ and $P_{D2}$ pigments. Thus, we assign the slight red shift as the hole migration towards a more delocalized cationic state, i.e., $Chl_{D1}{}^+Phe \longrightarrow (P_{D1}P_{D2})^+Phe$ (likely in addition to further environmental response to CS). Furthermore, the GSB band of exciton 1 and 2 exhibits blueshift from $1675\ cm^{-1}$ to $1678\ cm^{-1}$. This trend is consistent with the expectation that the lower frequency GSB of $Chl_{D1}$ ($1670\ cm^{-1}$), overlapping with the Phe ($1677\ cm^{-1}$), is replaced by the higher frequency band of P ($1682\ cm^{-1}$) following hole migration. Considering that the mode at $1713\ cm^{-1}$, the characteristic marker for $P^+$, only appears on a 6.3 ps timescale, it is very unlikely that $P^+$ contributes appreciably to the features at $1655\ cm^{-1}$ and $1666\ cm^{-1}$ at earlier waiting times. The evolution observed around $1655\ cm^{-1}$ and $1666\ cm^{-1}$ at later waiting times can therefore be understood as arising from both Phe$^-$ and $P^+$.

The final 41 ps component can be understood as predominantly reflecting CS limited by EET from peripheral Chlz to RC pigments as only significant evolution at the CS markers is observed on this timescale for exciton 8 (Fig. 4d). This timescale is also captured by the zero node line slope (ZNLS) present at $\omega_{det.} = 1670\ cm^{-1}$ (Fig. 5a, dotted line) in the spectra which decays with time constants (and amplitude) of $3.8 \pm 0.9$ ps (0.37) and $33 \pm 9$ ps (0.63) (Fig. 5b) and grossly indicates equilibration within the excitonic manifold. We note that while the ZNLS trends toward zero, a non-decaying component beyond the duration of the experiment (>100 ps) suggests the presence of inhomogeneous CS due to the different conformational distributions of the proteins on the ground state[21]. This timescale also falls within the previously established range (14 ps to 37 ps determined at temperatures of 77 K and 277 K, respectively) for EET from peripheral Chlz to RC pigments[13,19].

In summary, our results demonstrate that the CT state can be prepared directly upon photoexcitation, which is characterized by $Chl_{D1}{}^{\delta'+}Phe^{\delta'}$ ($\delta' > \delta$), and indicate that CS is facilitated by exciton-CT mixing with a contribution from $(Chl_{D1}{}^{\delta+}Phe^{\delta})^*$

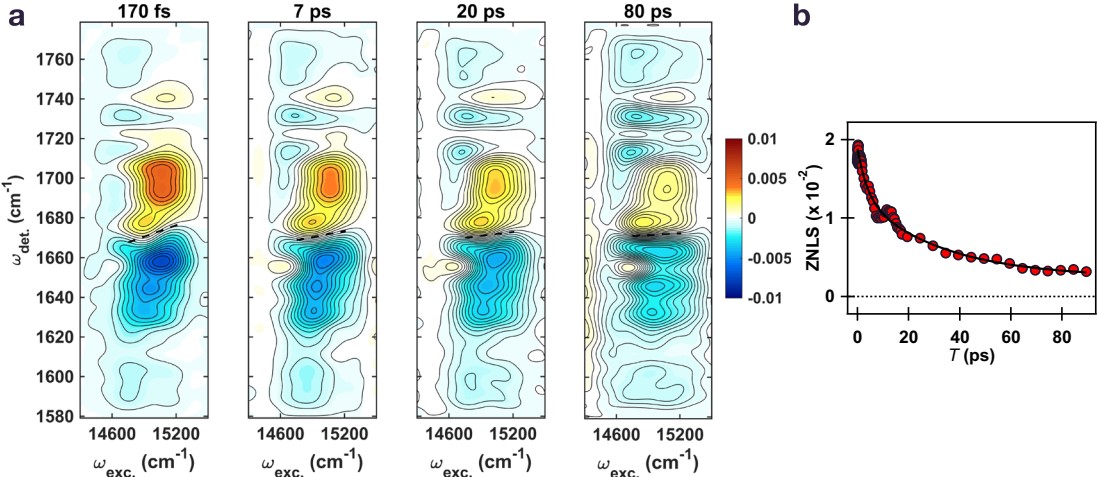

**Fig. 5 2DEV spectral evolution and ZNLS dynamics of the PSII-RC. a** 2DEV spectra of the PSII-RC at different waiting times. Zero node line slope (ZNLS), obtained by a linear fit of the zero signal intensity distribution along the excitation axis, is depicted in the spectra as a dotted line. Contour levels are drawn in 5% intervals. **b** ZNLS dynamics of the PSII-RC. Red dots indicate the ZNLS value at each waiting time and the black curve shows the fit result of double exponential functions (and an offset) with time constants of 3.8 ± 0.9 and 33 ± 9 ps.

throughout the excitonic manifold. The data further establish that there is no appreciable competition from $P_{D1}$—independent of excitation wavelength—indicating that the initial electron acceptor is Phe as supported by the observed vibrational structure at early waiting times. These results are entirely in agreement with the recent theoretical work of Sirohiwal et al. where the $Chl_{D1}{}^{+}$Phe CT state was found to be the lowest energy excitation within the PSII-RC with reasonable oscillator strength[41,52]. Further, no similarly low energy CT states involving $P_{D1}P_{D2}$ were found[41], thus theoretically excluding the special pair as a candidate for initial CS as our experimental data support. The dynamics indicate hole transfer occurs from $Chl_{D1}{}^{+}$ to $(P_{D1}P_{D2})^{+}$ on a 6.3 ps timescale as supported by the evolution of two distinct vibrational markers in order to minimize the potential influence from neighboring features. This is notably distinct from the bacterial RC where CS is largely initiated at the special pair (P) with the A branch bacteriochlorophyll acting as the primary acceptor. The distinct excitation asymmetry in the PSII-RC has been rationalized as a direct consequence of the electrostatic effect of the protein environment which likely arose as an evolutionary accommodation of water splitting in oxygenic photosynthetic systems (particularly its operation in the far-red)[39–41]. However, this remains an open question as the initial CS step itself in the PSII-RC has long evaded clear characterization.

## Methods

**Two-dimensional electronic-vibrational spectroscopy.** A detailed description of the experimental setup of 2DEV spectroscopy can be found elsewhere[43,48]. Briefly, the output of a Ti:sapphire oscillator (Vitara-S, Coherent) was regeneratively amplified with a 1 kHz repetition rate (Legend Elite, Coherent), an energy of 1 mJ/pulse, and a pulse duration of 40 fs. The amplified pulse was divided into two and one was used to pump a home-built visible non-collinear optical parametric amplifier (NOPA). The other pulse was used to generate a mid-IR probe pulse (centered at 5.9 μm) by difference frequency generation with signal and idler pulses from a near-IR collinear OPA. The output of the NOPA (centered at 675 nm, 60 nm fwhm) was compressed to 20 fs at the sample position using a pair of prisms and an acousto-optic dispersive programmable filter (AODPF, Dazzler, Fastlite). The pulse pair was introduced to a retroreflector on a motorized translation stage to control the waiting time, $T$, between the pump and probe pulses. The total power of the pump pulses was set at 80 nJ and the pulses were focused into the sample with spot size of 250 μm. The mid-IR pulse was divided by a 50:50 beam splitter to form probe and reference beams. The probe and reference beams were dispersed by a spectrometer (Horiba, Triax 180) and detected by a 64-pixel HgCdTe dual array (Infrared Systems Development). The cross-correlation between visible and mid-IR pulses was estimated to be 90 fs by a step-like transient IR response of a 50 μm Ge plate.

For each waiting time, a 2DEV spectrum was acquired by using the AODPF to scan the $t_1$ delay over 0–100 fs with 2.5 fs steps. For each $t_1$ delay, the signal was acquired with the relative phase between the pump pulses $\varphi_{12}$ set by 0, $2\pi/3$, and $4\pi/3$, and the desired signal was isolated by a $3 \times 1 \times 1$ phase cycling scheme[58,59]. The excitation axis was obtained by a Fourier transformation over $t_1$. The signal was collected in the fully rotated frame with respect to $t_1$. For 2D-EADS analysis, the detection range was selected to be $\omega_{det.} = 1620$–1740 cm$^{-1}$ because the dynamics in this range most fully reflect CS.

**Sample preparation.** All procedures for sample preparation were performed in the dark to minimize exposure to light as much as possible. We first isolated PSII-enriched membranes according to the previous literature with some modifications as follows[60,61]. We obtained spinach leaves (*Spinacia oleracea*) from a local store and kept in the dark overnight at 4 °C. The spinach leaves were briefly ground using a Waring blender in a buffer containing 50 mM MES-NaOH (pH 6.0), 400 mM NaCl, and 2 mM MgCl$_2$ at 4 °C. The ground tissues were filtered through four layers of Miracloth (Millipore), and the filtered homogenate was centrifuged at $1400 \times g$ for 10 min at 4 °C. The pellet was resuspended with a buffer containing 50 mM MES-NaOH (pH 6.0), 150 mM NaCl, and 5 mM MgCl$_2$, and resuspension was centrifuged at $4000 \times g$ for 10 min at 4 °C. The pellet was then resuspended with a buffer containing 50 mM MES-NaOH (pH 6.0), 15 mM NaCl, and 5 mM MgCl$_2$, and resuspension was centrifuged at $6000 \times g$ for 10 min at 4 °C. The pelleted thylakoid membranes were resuspended with the same buffer, and the concentration of chlorophylls was quantified by using 80% (v/v) acetone as described previously[62]. The thylakoid membranes (2.1 mg Chl/mL) were solubilized with 3.75% (w/v) Triton X-100 for 20 min on ice. The solution was centrifuged at $3500 \times g$ for 5 min at 4 °C. The supernatant was collected and further centrifuged at $40,000 \times g$ for 30 min at 4 °C. The pelleted PSII-enriched membranes were washed with the same buffer and centrifuged again at $40,000 \times g$ for 30 min at 4 °C. The PSII-enriched membranes were resuspended with a buffer containing 50 mM MES-NaOH (pH 6.0), 15 mM NaCl, 5 mM MgCl$_2$, and 400 mM sucrose, flash-frozen in liquid nitrogen, and stored at −80 °C until the following isolation procedures.

We isolated PSII-RC according to the previous literature with some modifications as follows[63–65]. The PSII-enriched membranes (1 mg Chl/mL) were solubilized with 4% (w/v) Triton X-100 in a buffer containing 50 mM Tris-HCl (pH 7.2) for 1 h on ice with gentle stirring. The solution was centrifuged at $33,000 \times g$ for 1 h at 4 °C. The supernatant was collected and loaded onto an anion exchange column (Toyopearl DEAE-650S resin) which was equilibrated with a buffer containing 50 mM Tris-HCl (pH 7.2), 30 mM NaCl, and 0.05% (w/v) Triton X-100 at 4 °C. The column was washed with the same buffer at a flow rate of 2.6 mL/min until the eluate showed the 417:435 nm ratio of about 1.16. Then, the column was subjected to a NaCl linear gradient from 30 to 200 mM at a flow rate of 1 mL/min. The green fraction eluted at 90–120 mM NaCl was collected. Then, the eluate was concentrated using Amicon centrifugal filters (10 K MWCO), spinning at $3200 \times g$ and 4 °C. The concentrated sample was diluted and re-concentrated with the buffer containing 50 mM Tris-HCl (pH 7.2), 0.4 M sucrose, 0.1% (w/v) n-dodecyl-β-D-maltoside (Anatrace) prepared with D$_2$O. The PSII-RC was flash-frozen and stored at −80 °C until 2DEV measurements.

For the spectroscopic experiments, the PSII-RC sample was mixed with glycerol-$d_8$ in an 80:20 (v/v) glycerol:PSII-RC ratio. The sample cell was constructed from two CaF$_2$ plates with a kapton spacer. The maximum optical density of the

PSII-RC sample in the investigated visible range was set at ~1.0 with a path length of 200 μm. The sample was placed in an optical cryostat (OptistatDN2, Oxford Instruments) at 77 K.

## Data availability

The data presented in this study are available from the corresponding author upon request.

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

## Acknowledgements

We thank Rafael Picorel for advice regarding isolation of the PSII-RC and Dr. Addison Schile for helpful comments on the manuscript. This research was supported by the U.S. Department of Energy, Office of Science, Basic Energy Sciences, Chemical Sciences, Geosciences, and Biosciences Division. Y.Y. appreciates the support of the Japan Society for the Promotion of Science (JSPS) Postdoctoral Fellowship for Research Abroad. E.A.A. acknowledges the support of the National Science Foundation Graduate Research Fellowship (Grant No. DGE 1752814).

## Author contributions

Y.Y. and G.R.F. conceived the research. Y.Y., E.A.A., S.-J.Y., and K.O. performed the 2DEV experiments. Y.Y. analyzed the experimental data. M.I. prepared the sample. Y.Y., E.A.A., and G.R.F. wrote the manuscript. All authors discussed the results and contributed to the manuscript.

## Competing interests

The authors declare no competing interests.
