## [Peer Review File · Nature Communications]

The initial charge separation step in oxygenic photosynthesisREVIEWER COMMENTS

Reviewer #1 (Remarks to the Author):

This paper applies a new experimental method to an important photosynthetic system, the photosystem II reaction center, to resolve questions about its charge separation mechanism. These are difficult experiments and the data interpretation is complicated. The authors interpret their results in the context of an excitonic model and claim to definitively determine that the initial charge separation begins with ChlD1+PheD1-, which can be directly excited optically. They rule out the possibility of an additional pathway involving PD1. The novelty of the method and the importance of the system make this paper appropriate for Nature Communications. However, the data analysis and interpretation as presented is difficult to follow. I suggest the following changes be made to aid in the readability of the paper. I also list inconsistencies that must be addressed to substantiate the claims made in the paper.

1) Figure 1 b should include panels that show the linear absorption (along the top) in addition to the pump (along the top) and probe spectra (along the side) used in the experiments. This is important since structured spectra can influence the resulting 2D EV spectra as well as the initial exciton population distribution.

2) Figure 2 should include a horizontal line to indicate the location of zero for the Phe-/Phe and P+/P difference spectra.

3) Figure 2 shows dashed/dotted lines that indicate features assigned to P+/P or Phe-/Phe. Additional lines should be added for other features that are discussed (such as 1750, 1764 cm⁻¹ assigned to ChlD1+). The same lines should be added to Figure 3a – several are conspicuously absent such as the 1701 cm⁻¹ line associated with P+/P. Figure 3a clearly shows a feature at 1701 cm⁻¹ associated with excitons 1 and 2 at 60 fs. Similarly, the features near 1700 cm⁻¹ in the 180 fs exciton 2 trace appear to correlate better with excitation of P+. The authors should explain how that can be consistent with their central claim that they observe a single charge separation pathway and exclude the possibility that charge separation is initiated at P.

4) The assignment lines should also be added along the detection axis to all of the 2D spectra that are shown in the paper as well as Figure 1 a-d in the SI.

5) Some of the language used to describe the spectral features is confusing – for example on page 7: “a clear dip at 1,657 cm⁻¹ appeared concomitantly with a new peak emerging at 1,666 cm⁻¹..”. These descriptions are reversed from the trends shown in the data.

6) The authors should describe what is shown in Figure 4 more clearly and add the assignment lines. Why didn't the authors use the 2D-EADS for this figure? Do the traces integrate over a range of excitation wavelengths?

Reviewer #2 (Remarks to the Author):

The paper entitled ‘The initial charge separation step in oxygenic photosynthesis’ by Yoneda et al. studied the charge separation and energy transfer processes of photosystem II-reaction center (PSII-RC) by two-dimensional electronic-vibrational spectroscopic (2DEV). The dynamics of RC consist of rapid energy transfer from high to low exciton states (<200 fs) and reversible charge transfer (mediated by the transient electric potential), a complex and versatile system guaranteeing its function to accomplish under various excitation conditions. Besides this character for most RC complex, the PSII-RC in plants is more difficult to fully understand, due to the severe spectral congestion of different cofactors. Thus, it is still far from unambiguity despite of extensive studies. This work mainly aimed to disentangle the two charge separation channels initiated at different pigment cofactor positions, (PD2+PD1⁻) or (ChlD1+PheD1⁻), utilizing 2DEV with advantage compared to other transient measurements. The application of 2DEV to the multi-channel multi-step dynamics of PSII-RC will provide additional and more detailed information for further understanding. However, I have questions about the analysis and the resulting conclusion that PD1PD2 was excluded for initiating charge separation as below, which should be addressed before reconsidering.

1. If charge separation initiates from PD2+PD1⁻, P⁺ should appear at later time when P+PheD1⁻ forms (the intermediate P+ChlD1⁻ is short-lived, because electron transfer from ChlD1 to PheD1 is relatively fast). In the text, only the characteristic P⁺/P spectrum was shown and discussed, but PD2+PD1⁻ or (PD2 δ +PD1 δ)^{*} were not. What is the characteristic spectrum for PD2+PD1⁻? Similar to that of P^{*} or P⁺/P? How to distinguish PD2+PD1⁻ with P⁺? Overall it is strange to me that the authors excluded PD2+PD1⁻ without discussing it.

2. There is an interesting phenomenon in the 2D-EADS as shown in Figure 4: while the 1704 cm⁻¹ band corresponding to the GSB keto mode of P decayed for all the exciton 1, 2, 5 and 8, the 1677 cm⁻¹ band corresponding to the GSB keto mode of Phe only decayed synchronously for exciton 5 and 8, but slightly increased in exciton 1 and nearly remained in exciton 2. At the same time, the CS

marker 1657 cm^{-1} band increased nearly synchronously. Doesn't it mean a correlation between P^* and CS marker?

3. One of the reasons for excluding PD_2+PD_1- is the band shift from 1716 to 1713 cm^{-1} at long time. However, the 3 cm^{-1} shift is more like the result of the decrease of the positive 1704 cm^{-1} band superposed on the 1716 cm^{-1} band. So I do not think it is reliable evidence.

4. A scheme may help to show the difference of the dynamic processes between this and other works.

Besides, there are some minor places to be checked:

Page 2: "the spectral resolution along the detection axis alone was not enough"

Page 13: "the latter which can be largely disentangled"

Page 14: the time constant values in Figure 4d are not consistent with those in the text.

Reviewer #3 (Remarks to the Author):

The authors present new experimental evidence to disentangle the long-standing discussion about the identity of the initial electron acceptor in the photosystem II-reaction center (PSII-RC) complex, which performs the initial light-energy conversion step in photosynthesis. The authors investigated the PSII-RC for the very first time with two-dimensional electronic-vibrational spectroscopy (2DEV), a technique which provides additional information with respect to previously applied "electronic-only" spectroscopy, especially crucial in spectrally congested systems as the PSII-RC, by addressing the vibrational modes of the different excited electronic states in a time-resolved manner. The spectroscopic data presented is of exceptional high quality, the interpretation is well supported by previously published data, and the conclusions are properly articulated.

In summary, I consider that this work is in line with the journal's high quality multidisciplinary research standards. I recommend this manuscript for publication in Nature Communications.

I only would like to rise the following points:

Page 5: "While uncertainty surrounds the involvement and extent of exciton-CT mixing in the PSII-RC". I consider that both the experimental (refs. 34, 22, 23) and theoretical work (refs. 35, and other articles by Novoderezhkin et al: Novoderezhkin, ChemPhysChem, 2011; Novoderezhkin, PCCP, 2015; and Novoderezhkin, PCCP, 2017) strongly indicate that the exciton states in the PSII RC are mixed with CT states. Therefore, I think that the statement "uncertainty surrounds the involvement and

extent of exciton-CT mixing in the PSII-RC" is not completely correct. I agree in that the identity of the exciton-CT states is under debate but not on the "involvement and extent" of such mixed states in the PSII-RC. I consider that this sentence should be rephrased.

Page 6: "For example, photoinduced absorptions (PIA) spanning $\omega_{det.} = 1,710-1,760 \text{ cm}^{-1}$ were seen to clearly favour the lower-lying excitonic states". I think this sentence is confusing. I suppose the authors mean that the PIA signal is stronger for the lower-lying excitonic states but I consider that the verb "favour" is not the most convenient one to describe the spectral features. I suggest changing this sentence for something like: "were seen to be more intense for the lower-lying excitonic states".

Page 9: "The corresponding peaks in the 2DEV spectrum (at $1,722 \text{ cm}^{-1}$, $1,730 \text{ cm}^{-1}$, and $1,740 \text{ cm}^{-1}$), apparent at early waiting times for exciton 2 and emerging later for exciton 8, are therefore assigned to Phe-." According to the slices of the 2DEV spectrum shown in figure 2 for excitons 2 and 8, I consider that both excitons 2 and 8 and both times (180 fs and 89 ps) show the $1,722 \text{ cm}^{-1}$, $1,730 \text{ cm}^{-1}$, and $1,740 \text{ cm}^{-1}$. The slight difference in band position in the intensity axis (higher for exciton 8) is most likely due to the overlap of the positive band centred around 1682 cm^{-1} . Could the authors comment on this point?

In addition, I have found two typos:

Page 5, line 2: "difference" should be "different"?

Figure 2 legend: "Vertical dotted (solid) lines indicate to band assignments corresponding P+/P" should be "Vertical dotted (solid) lines indicate band assignments corresponding to P+/P"?

We thank all the Reviewers for their thoughtful and constructive comments that have helped us to make significant improvements to the clarity of our paper. As one of the Reviewers pointed out, the features in 2DEV spectra are sensitive to the excitation spectrum and we wanted to eliminate any possible spectral distortion due to the structure of the excitation spectrum. Thus, we improved the shape of excitation laser spectrum and performed the experiment again. Now, the shape of the 2DEV spectral features is free from any unintended spectral structure caused by the excitation spectrum and the excitation wavelength dependence is now much more apparent than previously. In redoing these experiments, we also made improvements to the sample preparation method, which importantly leaves the RC protein more intact than our previous method. While the present results are slightly different from the previous data because of both the improved excitation spectrum and sample preparation method, we note that our conclusions remain the same. In the following, we provide a point-by-point response to all of the Reviewers' comments.

Reviewer #1 (Remarks to the Author):

This paper applies a new experimental method to an important photosynthetic system, the photosystem II reaction center, to resolve questions about its charge separation mechanism. These are difficult experiments and the data interpretation is complicated. The authors interpret their results in the context of an excitonic model and claim to definitively determine that the initial charge separation begins with ChlD1+PheD1-, which can be directly excited optically. They rule out the possibility of an additional pathway involving PD1. The novelty of the method and the importance of the system make this paper appropriate for Nature Communications. However, the data analysis and interpretation as presented is difficult to follow. I suggest the following changes be made to aid in the readability of the paper. I also list inconsistencies that must be addressed to substantiate the claims made in the paper.

1) Figure 1 b should include panels that show the linear absorption (along the top) in addition to the pump (along the top) and probe spectra (along the side) used in the experiments. This is important since structured spectra can influence the resulting 2D EV spectra as well as the initial exciton population distribution.

We feel that including the pump and probe spectra in Figure 1 would detract from our intended emphasis (the pigment structure of the RC and the excitation frequency-dependent vibrational structure) and overwhelm the figure. We do, however, agree that it is important to show the pump and probe spectra employed in this work. Thus, we added a new figure to the SI, Figure S1, which contains the linear absorption spectrum of the PSII-RC and pump and probe spectra of the laser pulses.

2) Figure 2 should include a horizontal line to indicate the location of zero for the Phe-/Phe and P+/P difference spectra.

Revised.

3) Figure 2 shows dashed/dotted lines that indicate features assigned to P+/P or Phe-/Phe. Additional lines should be added for other features that are discussed (such as 1750, 1764cm⁻¹ assigned to ChlD1+). The same lines should be added to Figure 3a – several are conspicuously absent such as the 1701 cm⁻¹ line associated with P+/P. Figure 3a clearly shows a feature at 1701 cm⁻¹ associated with excitons 1 and 2 at 60 fs. Similarly, the features near 1700cm⁻¹ in the 180fs exciton 2 trace appear to correlate better with excitation of P+. The authors should explain how that can be consistent with their central claim that they observe a single charge separation pathway and exclude the possibility that charge separation is initiated at P.

Our new spectra do not reliably reproduce the ω_{exc} dependence of the bands at 1,750 and 1,764 cm⁻¹, so we removed the discussion of these bands. However, we added a new discussion on the GSB feature around 1,677 cm⁻¹, which is related to Chl_{D1}. We added following sentences on page 10:

“the frequency of the GSB band around 1,675 cm⁻¹ for exciton 1 is lower than other excitonic states. This is in agreement with the previous reports that the GSB frequencies in this range of Chl_{D1} (1,670 cm⁻¹) and Phe (1,677 cm⁻¹) are redshifted compared to those of P (1,682 cm⁻¹) and Chlz (1,684 cm⁻¹).”

In Figure 3, we want to focus on the CT dynamics. In our spectra, a GSB can appear when the system is either on the excited state or in a CS state (for example, both P* and P⁺ show a GSB at 1,700 cm⁻¹). Because the excitonic states of PSII-RC are delocalized over several pigments and GSB/ESA modes reflect not only CS dynamics but also excitonic relaxation, we want to only focus on the assignment lines for specifically CS markers summarized as in the last paragraph of “General insights from the 2DEV spectra and IR band assignments” section on page 10. We also added following sentence on page 10:

“As the excitonic states of the PSII-RC are delocalized over several chromophores, we focus our discussion below on the CS markers rather than GSB and ESA features spanning 1,680-1,710 cm⁻¹ and 1,620-1,670 cm⁻¹, respectively, which reflect the relaxation of delocalized excitonic states.”

4) The assignment lines should also be added along the detection axis to all of the 2D spectra that are shown in the paper as well as Figure 1 a-d in the SI.

We thank the Reviewer for this point as we believe this will add clarity to our spectral analysis. In order not to overwhelm Figure 1b with both horizontal and vertical lines, we have added new a new figure to the SI, Figure S1b, which contains horizontal lines indicated IR assignments overlayed on the 2DEV spectrum from Figure 1b. Additionally, we have done the same for the 2D-EADS spectra in

Figure S2.

5) Some of the language used to describe the spectral features is confusing – for example on page 7: “a clear dip at 1,657 cm⁻¹ appeared concomitantly with a new peak emerging at 1,666 cm⁻¹..”. These descriptions are reversed from the trends shown in the data.

Thank you for pointing out our confusing wording. We revised the sentence on page 7 as follows: “Focusing on exciton 2, a clear GSB peak at 1,655 cm⁻¹, overlapping with a broad ESA feature, appeared concomitantly with a new peak emerging at 1,666 cm⁻¹.”

6) The authors should describe what is shown in Figure 4 more clearly and add the assignment lines. Our focus in Figure 4 is the red-shifting band at 1,713 cm⁻¹ of Chl⁺ and the appearance of the amide CO band at 1,657 cm⁻¹ (now in addition to the blue-shifting GSB band at 1,675 cm⁻¹—described in detail in the response to Reviewer 2’s third comment). In regards to adding assignment lines to Figure 4, we respectfully disagree. We would prefer not to have the figure overly congested to keep the reader’s attention on the features we described above.

Why didn't the authors use the 2D-EADS for this figure?

2D-EADS analysis is useful to provide averaged time constants and helpful to grasp general insight into the dynamics, however, the actual dynamics of the PSII-RC likely do not reflect a simple sequential process, as multiple parallel pathways are possible. So, we have instead used the 2D-EADS to guide what 2DEV slices are important to focus on. To address this point more explicitly in the manuscript, we have added following text on page 14:

“We note that the actual dynamics of the PSII-RC are not a simple sequential process as parallel and reversible processes are also expected. Thus, we discuss the dynamics based on the 2DEV slices rather than relying directly on the EADS.”

Do the traces integrate over a range of excitation wavelengths?

The trace of exciton 1 is integrated from $\omega_{exc.} = 14,500-14,650$ cm⁻¹ as in the case of Figure 3. To clarify this, we added following sentence to the caption of Figure 4:

“The energy ranges for $\omega_{exc.}$ are identical to those in Figure 3.”

Reviewer #2 (Remarks to the Author):

The paper entitled ‘The initial charge separation step in oxygenic photosynthesis’ by Yoneda et al. studied the charge separation and energy transfer processes of photosystem II-reaction center (PSII-RC) by two-dimensional electronic-vibrational spectroscopic (2DEV). The dynamics of RC consist of rapid energy transfer from high to low exciton states (<200 fs) and reversible charge transfer (mediated by the transient electric potential), a complex and versatile system guaranteeing its function to accomplish under various excitation conditions. Besides this character for most RC complex, the PSII-RC in plants is more difficult to fully understand, due to the severe spectral congestion of different cofactors. Thus, it is still far from unambiguity despite of extensive studies. This work mainly aimed to disentangle the two charge separation channels initiated at different pigment cofactor positions, (PD2+PD1⁻) or (ChlD1+PheD1⁻), utilizing 2DEV with advantage compared to other transient measurements. The application of 2DEV to the multi-channel multi-step dynamics of PSII-RC will provide additional and more detailed information for further understanding. However, I have questions about the analysis and the resulting conclusion that PD1PD2 was excluded for initiating charge separation as below, which should be addressed before reconsidering.

1. If charge separation initiates from PD2+PD1⁻, P⁺ should appear at later time when P+PheD1⁻ forms (the intermediate P+ChlD1⁻ is short-lived, because electron transfer from ChlD1 to PheD1 is relatively fast). In the text, only the characteristic P⁺/P spectrum was shown and discussed, but PD2+PD1⁻ or (PD2δ+PD1δ⁻)* were not. What is the characteristic spectrum for PD2+PD1⁻? Similar to that of P* or P⁺/P? How to distinguish PD2+PD1⁻ with P⁺? Overall it is strange to me that the authors excluded PD2+PD1⁻ without discussing it.

We appreciate the Reviewer for bringing this up—it is an important point. The spectra of P_{D2}⁺P_{D1}⁻ (and Chl_{D1}⁺ or Chl_{D1}⁻) are not accessible by steady state spectroscopy because these transients are short lived (if they indeed exist). The only way we can predict the P_{D2}⁺P_{D1}⁻ spectrum is to assume that cation and anion formation in P_{D2}P_{D1} will exhibit similar spectral shifts to monomeric Chl because the charges should be localized on P_{D2} and P_{D1}, respectively. The keto CO of the Chl⁻ generally red shifts compared to the ground state species (W. Mäntele et al., *Photochem. Photobiol.*, 1988, **47**, 451), however, this frequency falls in a congested region of the spectrum for the PSII-RC. On the other hand, the keto CO of monomeric Chl shows a ~25 cm⁻¹ blue shift upon cation formation (E. Nabedryk et al., *Biochemistry*, 1990, **29**, 3242). Given the main GSB peak of P is at 1,701 cm⁻¹, we can expect that the characteristic band of P_{D2}⁺P_{D1}⁻ should appear at ~1,726 cm⁻¹. However, as we only observe clear signatures of Phe⁻ bands (and associated GSBs) at 1,730 cm⁻¹ (and 1,722 and 1,740 cm⁻¹) across the entire excitation axis, we assign the initial electron acceptor as Phe—regardless of excitation wavelength. We added following sentence on page 10:

“It should be noted that the steady state spectrum of $P_{D2}^+P_{D1}^-$ is not measurable for a comparison because this species expected to be short lived (if it indeed exists as an intermediate). We therefore estimate the characteristic bands of $P_{D2}^+P_{D1}^-$ based on the assumption that cation and anion formation in $P_{D2}P_{D1}$ will exhibit similar spectral shifts to monomeric Chl because the charges should be localized on P_{D2} and P_{D1} , respectively. The keto CO of the Chl^- generally red shifts compared to the ground state species, however, this frequency falls in a congested region of the spectrum for the PSII-RC. On the other hand, the keto CO of monomeric Chl shows a $\sim 25\text{ cm}^{-1}$ blue shift upon cation formation. Given the main GSB peak of P is at $1,701\text{ cm}^{-1}$, we can expect that the characteristic band of $P_{D2}^+P_{D1}^-$ should appear at $\sim 1,726\text{ cm}^{-1}$. However, we only observe clear signatures of Phe^- bands (and associated GSBs) at $1,730\text{ cm}^{-1}$ (and $1,722$ and $1,740\text{ cm}^{-1}$) across the entire excitation axis.”

2. There is an interesting phenomenon in the 2D-EADS as shown in Figure 4: while the 1704 cm^{-1} band corresponding to the GSB keto mode of P decayed for all the exciton 1, 2, 5 and 8, the 1677 cm^{-1} band corresponding to the GSB keto mode of Phe only decayed synchronously for exciton 5 and 8, but slightly increased in exciton 1 and nearly remained in exciton 2. At the same time, the CS marker 1657 cm^{-1} band increased nearly synchronously. Doesn't it mean a correlation between P^* and CS marker?

We thank the Reviewer for their close inspection of the experimental results. The decay of the band at $1,704\text{ cm}^{-1}$ is due to *both* excitonic relaxation and charge separation. In this case, charge separation additionally leads to more pronounced signal intensity from the negative band at $1,705\text{ cm}^{-1}$ which is assigned to Phe^- and unfortunately directly cancels out the neighboring positive GSB band of P. We also note that it is difficult to distinguish excitonic relaxation from the charge separation dynamics based on GSB dynamics alone. So, we instead focus our discussion of the CS dynamics based on the specific markers of CS. We added following sentence on page 10:

“As the excitonic states of the PSII-RC are delocalized over several chromophores, we focus our discussion below on the CS markers rather than GSB and ESA features spanning $1680\text{-}1710\text{ cm}^{-1}$ and $1620\text{-}1670\text{ cm}^{-1}$, respectively, which reflect the relaxation of delocalized excitonic states.”

3. One of the reasons for excluding $P_{D2}^+P_{D1}^-$ is the band shift from 1716 to 1713 cm^{-1} at long time. However, the 3 cm^{-1} shift is more like the result of the decrease of the positive 1704 cm^{-1} band superposed on the 1716 cm^{-1} band. So I do not think it is reliable evidence.

We admit that the a few wavenumber shift may not be the not strongest evidence for the evolution from Chl_{D1}^+ to P^+ . In connection with this, we added a new paragraph as we described in the response to the comment 1 above. Briefly, as described in the response to comment 1, we expect the $P_{D2}^+P_{D1}^-$ signal to appear around $\sim 1,726\text{ cm}^{-1}$, however, we do not observe such a characteristic band in our

spectra. In addition, we observe a spectral shift at $\sim 1,675 \text{ cm}^{-1}$, which we also assigned to the hole transfer from Chl_{D1}^+ to P^+ . To address this, we added following sentence on page 16:

“Furthermore, the GSB band of exciton 1 and 2 undergoes a blueshift from $1,675 \text{ cm}^{-1}$ to $1,678 \text{ cm}^{-1}$. This trend is consistent with the expectation that the lower frequency GSB of Chl_{D1} ($1,670 \text{ cm}^{-1}$), overlapping with the Phe ($1,677 \text{ cm}^{-1}$), is replaced by the higher frequency band of P ($1,682 \text{ cm}^{-1}$) following hole migration.”

4. A scheme may help to show the difference of the dynamic processes between this and other works. We added a simplified scheme on Figure S2. However, we would also like to note that this scheme is not intended to imply that the dynamics of the PSII-RC follow a series of sequential, irreversible steps, but rather to grossly summarize the results of the 2D-EADS.

Besides, there are some minor places to be checked:

Page 2: “the spectral resolution along the detection axis alone was not enough

We revised the sentence as follows:

“only the spectral resolution along the detection axis was not enough to disentangle the distinct excitonic contributions”

Page 13: “the latter which can be largely disentangled”

We revised the sentence as follows:

“and the latter can be largely disentangled via vibrational signatures as we will show.”

Page 14: the time constant values in Figure 4d are not consistent with those in the text.

The time constants discussed in the text were obtained by global analysis and the time values shown in Figure 4 reflect particular waiting time. To make it clear, we added following sentence on page 14: “We note that the actual dynamics of the PSII-RC are not a simple sequential process as parallel and reversible processes are also expected. Thus, we discuss the dynamics based on the 2DEV slices rather than relying directly on the EADS.”

Reviewer #3 (Remarks to the Author):

The authors present new experimental evidence to disentangle the long-standing discussion about the identity of the initial electron acceptor in the photosystem II-reaction center (PSII-RC) complex, which performs the initial light-energy conversion step in photosynthesis. The authors investigated the PSII-RC for the very first time with two-dimensional electronic-vibrational spectroscopy (2DEV), a technique which provides additional information with respect to previously applied “electronic-only” spectroscopy, especially crucial in spectrally congested systems as the PSII-RC, by addressing the vibrational modes of the different excited electronic states in a time-resolved manner. The spectroscopic data presented is of exceptional high quality, the interpretation is well supported by previously published data, and the conclusions are properly articulated.

In summary, I consider that this work is in line with the journal’s high quality multidisciplinary research standards. I recommend this manuscript for publication in Nature Communications.

I only would like to rise the following points:

Page 5: “While uncertainty surrounds the involvement and extent of exciton-CT mixing in the PSII-RC”. I consider that both the experimental (refs. 34, 22, 23) and theoretical work (refs. 35, and other articles by Novoderezhkin et al: Novoderezhkin, ChemPhysChem,2011; Novoderezhkin, PCCP,2015; and Novoderezhkin, PCCP,2017) strongly indicate that the exciton states in the PSII RC are mixed with CT states. Therefore, I think that the statement “uncertainty surrounds the involvement and extent of exciton-CT mixing in the PSII-RC” is not completely correct. I agree in that the identity of the exciton-CT states is under debate but not on the “involvement and extent” of such mixed states in the PSII-RC. I consider that this sentence should be rephrased.

The Reviewer makes a good point and we appreciate the many suggested references. We have revised the sentence as follows and added the suggested references:

“While uncertainty surrounds the identity of the states involved in exciton-CT mixing in the PSII-RC,”

Page 6: “For example, photoinduced absorptions (PIA) spanning $\omega_{\text{det.}} = 1,710\text{-}1,760\text{ cm}^{-1}$ were seen to clearly favour the lower-lying excitonic states”. I think this sentence is confusing. I suppose the authors mean that the PIA signal is stronger for the lower-lying excitonic states but I consider that the verb “favour” is not the most convenient one to describe the spectral features. I suggest changing this sentence for something like: “were seen to be more intense for the lower-lying excitonic states”.

We revised the text as suggested by the Reviewer.

Page 9: “The corresponding peaks in the 2DEV spectrum (at $1,722\text{ cm}^{-1}$, $1,730\text{ cm}^{-1}$, and $1,740\text{ cm}^{-1}$), apparent at early waiting times for exciton 2 and emerging later for exciton 8, are therefore assigned

to Phe-.” According to the slices of the 2DEV spectrum shown in figure 2 for excitons 2 and 8, I consider that both excitons 2 and 8 and both times (180 fs and 89 ps) show the 1,722 cm⁻¹, 1,730 cm⁻¹, and 1,740 cm⁻¹. The slight difference in band position in the intensity axis (higher for excitons 8) is most likely due to the overlap of the positive band centred around 1682 cm⁻¹. Could the authors comment on this point?

The width of the vibrational band of this region (mainly keto and ester CO) is generally less than 10 cm⁻¹ (for example, R. Wang, et al., *Biophys. J.*, 2004, 86, 1061), so we do not expect overlap from the positive band at ~1,680 cm⁻¹ to influence the higher frequency region around > 1,700 cm⁻¹. We attribute the appearance of the GSB bands at 1,730 cm⁻¹ and 1,740 cm⁻¹ for exciton 8 at 180 fs to overlap with lower-lying excitonic states. To clarify this, we revised the text on page 9 as follows:

“It should be noted that exciton 8 does show a small negative signal around 1,730 cm⁻¹ and a positive band at 1,740 cm⁻¹ immediately after photoexcitation, despite being characterized by ChlZ_{D1}. We attribute these signals to either small contributions from the ester ESA or some degree of overlap between excitonic bands, as these slices only represent the calculated zero phonon transitions and the actual absorption has finite bandwidth.”

In addition, I have found two typos:

Page 5, line 2: “difference” should be “different”?

We thank the Reviewer for catching this typo.

Figure 2 legend: “Vertical dotted (solid) lines indicate to band assignments corresponding P+/P” should be “Vertical dotted (solid) lines indicate band assignments corresponding to P+/P”?

We thank the Reviewer for catching this typo.

REVIEWER COMMENTS

Reviewer #1 (Remarks to the Author):

The authors have addressed the majority of concerns and have improved the paper. I have several remaining concerns:

1) The authors use both PIA and ESA which is confusing

2) Although 2DEV clearly offers some advantages, the authors are overselling its capabilities compared to previous work. On page 12 the paper claims “The lack of P+ is in contrast to several previous spectroscopic studies that suggested there are two CS pathways in the PSII-RC.21,22,24,38 However, these experiments utilized spectroscopic methods solely in the visible region which are significantly disadvantaged when it comes to untangling the highly overlapping signals of the relevant states. ” While it is true that the Qy region contains highly overlapping states, other regions of the visible spectrum such as the Qx region that allow for clearer separation of the contributions as demonstrated in reference 21. Note that although they did not use 2D spectroscopy in reference 21, they tuned the pump pulse to excite different initial states in a similar manner.

3) Regarding point 3) in my original review:

“Figure 2 shows dashed/dotted lines that indicate features assigned to P+/P or Phe-/Phe. Additional lines should be added for other features that are discussed (such as 1750, 1764cm⁻¹ assigned to ChlD1+). The same lines should be added to Figure 3a – several are conspicuously absent such as the 1701 cm⁻¹ line associated with P+/P. Figure 3a clearly shows a feature at 1701 cm⁻¹ associated with excitons 1 and 2 at 60 fs. Similarly, the features near 1700cm⁻¹ in the 180fs exciton 2 trace appear to correlate better with excitation of P+. The authors should explain how that can be consistent with their central claim that they observe a single charge separation pathway and exclude the possibility that charge separation is initiated at P.”

The authors give the following reply:

“In Figure 3, we want to focus on the CT dynamics. In our spectra, a GSB can appear when the system is either on the excited state or in a CS state (for example, both P* and P+ show a GSB at 1,700 cm⁻¹). Because the excitonic states of PSII-RC are delocalized over several pigments and GSB/ESA modes reflect not only CS dynamics but also excitonic relaxation, we want to only focus on the assignment lines for specifically CS markers summarized as in the last paragraph of “General insights from the 2DEV spectra and IR band assignments” section on page 10. We also added following sentence on page 10:

“As the excitonic states of the PSII-RC are delocalized over several chromophores, we focus our discussion below on the CS markers rather than GSB and ESA features spanning 1,680-1,710 cm⁻¹ and 1,620-1,670 cm⁻¹, respectively, which reflect the relaxation of delocalized excitonic states.” ”

In Figure 2, the authors draw lines to indicate features they will use as CS markers, indicating 3 markers for P+ (dotted lines). However, as they explain above (but not clearly in the manuscript), they discount the use of 2 of these 3 markers, leaving only one (weak) marker at 1713 cm⁻¹ for identifying the P+ state, compared to 4 markers for identifying the Phe- state. They should modify Figure 2 to make it clear which markers they are actually using and why (or perhaps include a table with the assignments and supporting references to make it easier for the reader). They should also explicitly state the difficulty they face in observing the P+ pathway with this experiment since this directly impacts the central claim of the paper.

Reviewer #2 (Remarks to the Author):

The revised version of the manuscript addresses my concern acceptably, so should be published in Nature Communications.

Reviewer #3 (Remarks to the Author):

I consider that the authors have addressed the reviewer's comments satisfactorily. The new data presented taken with an improved excitation laser spectrum shape is highly appreciated as well as the clarifications in the figures and in the manuscript text. Therefore, I recommend this manuscript for publication in Nature Communications.

I only want to point out a doubt I have about the last sentence of the manuscript:

"However, this remains an open question as the initial CS step itself in the has long evaded clear characterization."

I believe that there may be some typo or missing word (maybe "PSII-RC"?) after "CS step itself in the".

REVIEWER COMMENTS

Reviewer #1 (Remarks to the Author):

The authors have addressed the majority of concerns and have improved the paper. I have several remaining concerns:

1) The authors use both PIA and ESA which is confusing.

We apologize for the confusion. We use ESA to indicate absorption from excited states and PIA to indicate absorption from newly formed species (which may or may not be in an excited state). To clarify, we added the following on page 6:

“For example, absorption from newly formed species (photoinduced absorptions, PIA), ...”

2) Although 2DEV clearly offers some advantages, the authors are overselling its capabilities compared to previous work. On page 12 the paper claims “The lack of P+ is in contrast to several previous spectroscopic studies that suggested there are two CS pathways in the PSII-RC.21,22,24,38 However, these experiments utilized spectroscopic methods solely in the visible region which are significantly disadvantaged when it comes to untangling the highly overlapping signals of the relevant states.” While it is true that the Qy region contains highly overlapping states, other regions of the visible spectrum such as the Qx region that allow for clearer separation of the contributions as demonstrated in reference 21. Note that although they did not use 2D spectroscopy in reference 21, they tuned the pump pulse to excite different initial states in a similar manner.

We did not intend to minimize the earlier contributions of others and apologize if Reviewer 1 felt this was our intention. However, 2DEV spectroscopy is different from pump-probe techniques: signals arise from a conditional probability such that excitation at wavelength X leads to vibrational transition(s) at frequency Y. This conditional probability lacks the convolution over pump frequencies that is inevitable in pump-probe spectroscopies. To avoid impugning prior work, we have reworded the second and third sentences on page 13 to read:

“However, the resolution afforded by both the visible excitation and IR detection dimensions of the 2DEV spectrum lead in particular to the vibrational characterization of exciton 1—providing direct evidence that the initial CT...”

3) Regarding point 3) in my original review:

“Figure 2 shows dashed/dotted lines that indicate features assigned to P+/P or Phe-/Phe. Additional lines should be added for other features that are discussed (such as 1750, 1764cm⁻¹ assigned to ChlD1+). The same lines should be added to Figure 3a – several are conspicuously absent such as the 1701 cm⁻¹ line associated with P+/P. Figure 3a clearly shows a feature at 1701 cm⁻¹ associated with excitons 1 and 2 at 60 fs. Similarly, the features near 1700cm⁻¹ in the 180fs exciton 2 trace appear to correlate better with excitation of P+. The authors should explain how that can be consistent with their central claim that they observe a single charge separation pathway and exclude the possibility that charge separation is initiated at P.”

The authors give the following reply:

“In Figure 3, we want to focus on the CT dynamics. In our spectra, a GSB can appear when the system is either on the excited state or in a CS state (for example, both P* and P+ show a GSB at 1,700 cm⁻¹). Because the excitonic states of PSII-RC are delocalized over several pigments and GSB/ESA modes reflect not only CS dynamics but also excitonic relaxation, we want to only focus on the assignment lines for specifically CS markers summarized as in the last paragraph of “General insights from the 2DEV spectra and IR band assignments” section on page 10. We also added following sentence on page 10:

“As the excitonic states of the PSII-RC are delocalized over several chromophores, we focus our discussion below on the CS markers rather than GSB and ESA features spanning 1,680-1,710 cm⁻¹ and 1,620-1,670 cm⁻¹, respectively, which reflect the relaxation of delocalized excitonic states.” ”

In Figure 2, the authors draw lines to indicate features they will use as CS markers, indicating 3 markers for P+ (dotted lines). However, as they explain above (but not clearly in the manuscript), they discount the use of 2 of these 3 markers, leaving only one (weak) marker at 1713 cm⁻¹ for identifying the P+ state, compared to 4 markers for identifying the Phe- state. They should modify Figure 2 to make it clear which markers they are actually using and why (or perhaps include a table with the assignments and supporting references to make it easier for the reader).

In the previous revision, we added an explanation of why we used the assignment lines shown in Figure 3 to track CS dynamics in the last paragraph of the “General insights from the 2DEV spectra and IR band assignments” section on page 10. However, as the Reviewer pointed out, a table which summarizes the assignments (and provides further clarification for the assignment lines in Figure 3) will likely be useful for a broad readership. We added the following table on page 11:

Table 1. IR frequency assignments of the PSII-RC. The markers used to track the CS dynamics are shown in bold.

Frequency (cm ⁻¹)	Assignment
1655	GSB of amide CO^{11,31}
1666	PIA of amide CO^{11,31}
1677	GSB of Phe ^{29,32}
1682	GSB of P ^{31,32}
1705	GSB of P ^{31,32}
1713	PIA of P^{+31,32}
1722	GSB of Phe or amide CO^{29,32}
1730	PIA of Phe⁻ or amide CO^{29,32}
1740	GSB of Phe or amide CO^{29,32}

They should also explicitly state the difficulty they face in observing the P⁺ pathway with this experiment since this directly impacts the central claim of the paper.

The data establish that there is no appreciable contribution from P⁺ at the earliest resolvable waiting time. We have clarified this finding in our concluding comments. In regards to establishing the formation of P⁺ at later waiting times, we discussed this in detail in our expanded discussion (in our previous revision) of the dynamics of several key markers for hole transfer from Chl⁺ to P⁺ (see pages 16 and 17 in the manuscript). We changed the text in the concluding comments section as follows:

“The data further establish that there is no appreciable competition from P_{D1}—independent of excitation wavelength—indicating that the initial electron acceptor is Phe as supported by the observed vibrational structure at early waiting times. The dynamics indicate hole transfer occurs from Chl_{D1}⁺ to (P_{D1}P_{D2})⁺ on a 6.3 ps timescale as supported by the evolution of two distinct vibrational markers in order to minimize the potential influence from neighboring features.”

Reviewer #2 (Remarks to the Author):

The revised version of the manuscript addresses my concern acceptably, so should be published in Nature Communications.

Thank you.

Reviewer #3 (Remarks to the Author):

I consider that the authors have addressed the reviewer's comments satisfactorily. The new data presented taken with an improved excitation laser spectrum shape is highly appreciated as well as the clarifications in the figures and in the manuscript text. Therefore, I recommend this manuscript for publication in Nature Communications.

I only want to point out a doubt I have about the last sentence of the manuscript:

“However, this remains an open question as the initial CS step itself in the has long evaded clear characterization.”

I believe that there may be some typo or missing word (maybe "PSII-RC"?) after “CS step itself in the”.

We revised the sentence. Thank you.

REVIEWERS' COMMENTS

Reviewer #1 (Remarks to the Author):

In their rebuttal, the authors claim “The data establish that there is no appreciable contribution from P+ at the earliest resolvable waiting time”. It appears that this entire claim that is central to the paper rests on a small shift of one peak from 1716cm⁻¹ (assigned to ChD1+) to 1713cm⁻¹ (assigned to P+, although on page 10 of the paper P+ is assigned to 1711cm⁻¹) shown in Figure 4. The authors should add vertical lines at 1713cm⁻¹ and 1716cm⁻¹ to allow this shift to be more easily evaluated. In contrast to the claim of the authors, for Exciton 1 the shift to 1713cm⁻¹ appears to be complete by 1.3ps, which undermines the claim that P+ appears only at considerably longer times. For excitons 5 and 8, the competing positive signal in the spectral region of the 1713cm⁻¹ peak makes it difficult to draw any substantive conclusions about a frequency shift.

In SI Figure S2b, c and d, the authors should include axes on the insets. It would also be helpful to include vertical lines showing the frequencies of the CS markers given in Table 1 in this figure as well as Figure 4 of the main text.

Reviewer #2 (Remarks to the Author):

I think the concerns from the Reviewer 1 are significant for improving the manuscript, rather than point to fatal errors. Considering the author's reply, I think the present version could be accepted by Nature Communications.

REVIEWER COMMENTS

Reviewer #1 (Remarks to the Author):

In their rebuttal, the authors claim “The data establish that there is no appreciable contribution from P⁺ at the earliest resolvable waiting time”. It appears that this entire claim that is central to the paper rests on a small shift of one peak from 1716cm⁻¹ (assigned to ChD1⁺) to 1713cm⁻¹ (assigned to P⁺, although on page 10 of the paper P⁺ is assigned to 1711cm⁻¹) shown in Figure 4. The authors should add vertical lines at 1713cm⁻¹ and 1716cm⁻¹ to allow this shift to be more easily evaluated.

Our conclusions on the contribution of P⁺ at early time are based on both the red-shift of the band at 1,713 cm⁻¹ and the blue-shift of the Chl ground state bleach at 1,677 cm⁻¹, as we replied in the first revision. Regarding the vertical lines we experimented with one line at 1,713 cm⁻¹, and up to six lines at 1,655, 1,666, 1,713, 1,722, 1,730 and 1,740 cm⁻¹. The spectra are difficult to see with more than one line and we have replaced Fig 4 with spectra containing a single line at 1,713 cm⁻¹. We hope this improves clarity of the shift we describe.

In contrast to the claim of the authors, for Exciton 1 the shift to 1713cm⁻¹ appears to be complete by 1.3ps, which undermines the claim that P⁺ appears only at considerably longer times.

Regarding the time scale of the red-shifting of exciton 1, the band at ~1,716 cm⁻¹ indeed shows a small shift within 1.3 ps, however, completion of the shift to 1,713 cm⁻¹ can only be seen at 10 picoseconds or later, indicating that the hole transfer from Chl_{D1}⁺ to P⁺ occurs on a picosecond time scale. This picture is completely consistent with two recent theory/modeling papers (H. Tamura, et al., *Chem. Sci.*, 2021; A. Sirohiwal and D. A. Pantazis, *Angew. Chemie Int. Ed.*, 2022). We have added these two references to the paper. However, it is difficult to completely rule out the contribution of P⁺ at early time, because the two bands of Chl_{D1}⁺ and P⁺ overlap. The two recent theoretical studies suggest that the lowest CT state among the RC pigments is Phe⁻P⁺ and that state can also be directly photo excited at 730 nm-750 nm with, however, very low oscillator strength. In contrast the Chl_{D1}⁺Phe⁻ transition has significant oscillator strength as a result of its mixed excitonic character (Sirohiwal et al). We added the following text on page 10.

“Recent theoretical studies suggest that the lowest CT state among the RC pigments is composed of P⁺Phe⁻ and that state, which has very low oscillator strength, can be directly excited by far-red light (in the red tail of, or beyond our laser spectrum). Our spectra show similar frequencies for Chl_{D1}⁺ and P⁺, thus it is possible that there is a small contribution from P⁺ to the signal even at early time. It is clear, however, that the majority of the initial signal at 1716 cm⁻¹ and 1,677 cm⁻¹ arises from Chl_{D1} because of the significant oscillator strength of Chl_{D1}⁺Phe⁻ transition in addition to indicating that the initial electron acceptor is Phe.”

We also changed “P⁺” to “P_{D2}⁺ P_{D1}⁻” on page 10 to be consistent with the above argument.

For excitons 5 and 8, the competing positive signal in the spectral region of the 1713cm⁻¹ peak makes it difficult to draw any substantive conclusions about a frequency shift.

For exciton 5 and 8, although the positive signal overlap with the 1713 cm⁻¹ band, it clearly shows red-shifting behavior and we don't see any significant contribution from the P_{D1}⁻ P_{D2}⁺ band. Thus, we can conclude that there is no appreciable competition from P_{D1}⁻ P_{D2}⁺—independent of excitation wavelength.

In SI Figure S2 b, c and d, the authors should include axes on the insets. It would also be helpful to include vertical lines showing the frequencies of the CS markers given in Table 1 in this figure as well as Figure 4 of the main text.

We added vertical lines showing CS markers at 1,713 cm⁻¹ (and 1,722 cm⁻¹) for the insets of Figure S2. Regarding Figure 4, please see the reply for the first comment.